# Minoxidil Nanoparticles Targeting Hair Follicles Enhance Hair Growth in C57BL/6 Mice

**DOI:** 10.3390/pharmaceutics14050947

**Published:** 2022-04-27

**Authors:** Yoshihiro Oaku, Akinari Abe, Yohei Sasano, Fuka Sasaki, Chika Kubota, Naoki Yamamoto, Tohru Nagahama, Noriaki Nagai

**Affiliations:** 1Research & Development Headquarters Self-Medication, Taisho Pharmaceutical Co., Ltd., 1-403 Yoshinocho, Saitama 331-9530, Japan; y-oaku@taisho.co.jp (Y.O.); a-abe@taisho.co.jp (A.A.); t-nagahama@taisho.co.jp (T.N.); 2Faculty of Pharmacy, Kindai University, 3-4-1 Kowakae, Higashiosaka 577-8502, Japan; 1611610034b@kindai.ac.jp (Y.S.); 1811610014f@kindai.ac.jp (F.S.); kubota.chika@kindai.ac.jp (C.K.); 3Research Promotion and Support Headquarters, Center for Clinical Trial and Research Support, Fujita Health University, 1-98 Dengakugakubo, Toyoake 470-1192, Japan; naokiy@fujita-hu.ac.jp

**Keywords:** minoxidil, nanoparticle, hair follicle delivery, hair growth, hair follicle epithelial stem cell, androgenetic alopecia

## Abstract

We previously found that 1% minoxidil (MXD) nanoparticles prepared using a bead mill method led to an increase I n hair follicle delivery and hair growth in C57BL/6 mice. In the present study, we designed a nanoparticle formulation containing 5% MXD (MXD-NPs) using the bead mill method and investigated the hair-growth effect of MXD-NPs and a commercially available MXD solution (CA-MXD). Hair growth and in vivo permeation studies were conducted using C57BL/6 mice. Moreover, we examined the MXD contents in the upper (hair bulge) and the lower hair follicle (hair bulb) and observed the hair follicle epithelial stem cells (HFSC) by immunohistochemical staining using the CD200 antibody. The mean particle size of the MXD in the MXD-NPs was 139.8 nm ± 8.9 nm. The hair-growth effect of the MXD-NPs was higher than that of CA-MXD, and the MXD content in the hair bulge of mice treated with MXD-NPs was 7.4-fold that of the mice treated with CA-MXD. In addition, the activation of HFSC was observed around the bulge in the MXD-NPs-treated mice. We showed that MXD-NPs enable the accumulation of MXD in the upper hair follicles more efficiently than CA-MXD, leading the activation of HFSC and the hair growth.

## 1. Introduction

Androgenetic alopecia (AGA) is an age-dependent disorder resulting in patterned hair loss. In particular, 30% of men will have developed AGA at the age of 30 years, and 50% of men will have developed AGA at the age of 50 years [1]. The topical application of minoxidil (MXD) is a first-line therapy for AGA [2]. Only MXD is approved by the U.S. Food and Drug Administration (FDA) as a topical solution. A five-percent MXD solution is the highest concentration available among products that have been approved by the FDA [3]. The clinical efficacy of MXD treatment for AGA has been reported, and the efficacy of 5% MXD is significantly higher than that of 2% or 1% MXD [4,5]. According to these studies [4,5], some people have been dissatisfied with the efficacy of the 5% MXD treatments. Clinical studies on the efficacy of 10% MXD treatments for AGA were performed to see whether 10% MXD treatment could provide the expected clinical outcomes; however, no significant advantage was observed for the 10% MXD treatment, compared with the 5% MXD treatment [6,7]. In contrast, the incidence of adverse events associated with the 10% MXD treatment was higher than that of the 5% MXD treatment. Therefore, novel approaches to achieve both the expected efficacy and safety are needed.

MXD is known to act mainly on the hair cycle and to stimulate hair growth [8]. Studies using stump-tailed macaques showed that MXD maintains the anagen phase and delays the initiation of the telogen phase in the hair cycle [9,10]. Based on these actions of MXD on the hair cycle, nanotechnologies have been introduced as potential technologies that can deliver MXD into hair follicles [11,12], such as solid nanoparticles [13], polymer nanoparticles [14,15,16], nanoemulsions [17,18] and nanovesicles [19,20,21]. Oliveira et al. have reported nanostructured lipid carriers (NLC) loaded with minoxidil sulphate, an active metabolite of MXD, and the NLC was deposited into hair follicles [22]. Gelfuso et al. reported that the combination with minoxidil sulphate-loaded microparticles and iontophoresis improved the follicular delivery [23]. In addition, Kim et al. showed that minoxidil-loaded microneedles were effective for the hair growth in mice [24]. We previously designed a 1% MXD solid nanoparticle using the bead mill method, and these MXD solid nanoparticles increased hair growth in C57BL/6 mice through the accumulation of MXD in hair follicles [25]. Thus, MXD solid nanoparticles may enhance the therapeutic effects of MXD for AGA, improving patient satisfaction in terms of efficacy and safety. Nanotechnologies that can enhance the targeting of MXD into hair follicles are reported [14,15,16,17,18,19]; however, the MXD contents in these formulations are less than 5%. On the other hand, 5% MXD formulations have been developed [13,20,21]; the design of 5% MXD formulations exhibiting both the dispersibility of MXD nanoparticles and the enhancement of hair-growth effects through the use of nanoparticles that target hair follicles has not been accomplished. In the present study, we used the bead mill method to prepare a 5% MXD nanoparticle formulation (MXD-NPs) with the aim of achieving both the maintenance of the dispersion state and the enhancement of hair-growth effects.

In evaluating the hair-growth effects of MXD, the selection of appropriate animal models and hair-growth factors is important. Other than humans, the stump-tailed macaque is known as an animal that develops AGA and is used as an animal model to translate hair-growth effects to humans [10]. Besides, hair-growth test using mice, observation of human hair follicles organ and proliferation analysis of human dermal papilla cells is conducted from the viewpoint of the convenience for evaluating hair-growth effects [26]. On the other hand, there are differences between human and mice hair cycle: the human hair cycle is mosaic, while the mice hair cycle is synchronized [27]. In regard to MXD, the hair-growth efficacy and the concentration dependency are verified in human [4], stump-tailed macaque [9] and mice [28]. In this regard, we selected mice as an animal model. The C57BL/6 mouse is an animal model for evaluating hair-growth effects, and these mice have been widely used to determine the hair-growth effects of formulations [20]. On the other hand, during the process of hair growth, MXD is known to enhance the expression of hair-growth factors, including both an insulin-like growth factor-1 (IGF-1) and a vascular endothelial growth factor (VEGF) [29,30]. In addition, the hair follicle epithelial stem cells in the bulge region (detected in the outer root sheath on the hair root side) are strongly associated with hair matrix cell development. Reportedly, CD200, cytokeratin 15 and CD34 can be used as hair follicle epithelial stem cell markers [31]. In particular, CD200 is expressed around the bulge of hair follicles and on the outer sheath on the root side; consequently, the proliferation and activation of hair follicle epithelial stem cells in the bulge region was evaluated using immunohistochemical staining with CD 200 antibody in the present study.

Based on the results of the previous study [25], we designed a nanoparticle formulation containing a high MXD content (5% MXD). We investigated whether the hair-growth effect of MXD-NPs is higher than that of a commercially available MXD solution (CA-MXD) using C57BL/6 mice and examined the effect of MXD-NPs on the production of IGF-1 and VEGF. In addition, we newly investigated whether MXD is targeted in the upper part (hair bulge) or lower part (hair bulb) of the hair follicles, and the activation of the hair follicle epithelial stem cells using CD200.

## 2. Materials and Methods

### 2.1. Chemicals

All other chemicals used were of the highest purity commercially available. Briefly, the CA-MXD (Riup 5%) and MXD powder were supplied by Taisho Pharmaceutical Co., Ltd. (Tokyo, Japan), and TRIzol reagent was obtained from Life Technologies Inc. (Rockville, MD, USA). Methyl cellulose (MC) was provided by Shin-Etsu Chemical Co., Ltd. (Tokyo, Japan). Propyl p-hydroxybenzoate, Mannitol and methyl p-hydroxybenzoate were purchased from Wako Pure Chemical Industries, Ltd. (Osaka, Japan). RNA PCR Kit (AMV Ver 3.0) was obtained from TaKaRa Bio Inc. (Shiga, Japan). LightCycler FastStart DNA Master SYBR Green I and a Bio-Rad Protein Assay Kit were purchased from Roche Diagnostics Applied Science (Mannheim, Germany) and Bio-Rad Laboratories (Hercules, CA, USA), respectively. A Mouse/Rat insulin-like growth factor-1 (IGF-1) ELISA kit and vascular endothelial growth factor (VEGF) ELISA kit were purchased from R&D Systems (Minneapolis, MN, USA) and Cosmo Bio Co., Ltd. (Tokyo, Japan), respectively.

### 2.2. Animals

C57BL/6 mice aged 7 weeks old (male) were obtained from Clea Japan, Inc. (Tokyo, Japan), and the back hair was removed before use in the experiments to evaluate hair follicle (hair bulge and bulb region) delivery and the hair-growth effect of MXD. All C57BL/6 mice were housed under normal conditions and fed a CE-2 formulation diet (Clea Japan Inc., Tokyo, Japan). The MXD formulation (30 µL) was applied repetitively to the dorsal area (2 cm^2^) of C57BL/6 mice once a day (10:30 a.m.). These experiments using C57BL/6 mice were approved on 1 April 2017 by Kindai University (project identification code KAPS-29-007), and were performed in accordance with the guidelines of Kindai University.

### 2.3. Preparation of MXD Formulations

A dispersion containing MXD nanoparticles was prepared by referring to our previous reports [25,32,33]. MXD powder and MC were mixed and then milled using an agate mortar under cold conditions (4 °C) for 1.5 h, and the mixture (MXD, 5 g and MC, 8 g) was suspended in 100 mL of purified water containing methyl p-hydroxybenzoate (0.026%) and propyl p-hydroxybenzoate (0.014%) as preservatives); the dispersion was then stirred in 2-mL tubes with zirconia beads (diameter, 0.1 mm). After that, the mixture was milled by the combination of ultrasonic treatment for 5 min (W-113MK-II, Honda Electronics Co., Ltd., Aichi, Japan) and Micro Smash MS-100R at 5500 rpm for 30 s (TOMY DIGITAL BIOLOGY Co., Ltd., Tokyo, Japan). The ultrasonic and bead mill treatments were alternately repeated 30 times at 4 °C. Furthermore, the milled dispersion was crushed using a ShakeMaster^®^ NEO (Bio Medical Science, Tokyo, Japan) with zirconia beads (diameter, 0.1 mm) for 120 min at 1500 rpm, and the obtained dispersions were used as MXD-NPs. Non-milled dispersions with the same composition (5% MXD, 8% MC, 0.026% methyl p-hydroxybenzoate and 0.014% propyl p-hydroxybenzoate in purified water, *w/v* %) were defined as MXD-MPs (control), and the pH of both MXD formulations was adjusted to 6.8. In the present study, MC was added to enhance the dispersibility of the MXD nanoparticles and to increase the crushing efficiency in the bead mill [25].

### 2.4. Measurement of MXD Levels

For analysis of the MXD level, an LC-20AT system (HPLC; Shimadzu Corp., Kyoto, Japan) was used with an Inertsil^®^ ODS-3 column (GL Science Co., Inc., Tokyo, Japan) at 35 °C. One microgram per milliliter of ethyl p-hydroxybenzoate was used as an internal standard; 50 µL of sample and 100 µL of methanol containing 0.1 µg of ethyl p-hydroxybenzoate were mixed, and 10 μL of the mixture were injected. Methanol/purified water containing 3 mM docusate sodium (1/1, *v*/*v*) was used as the mobile phase, followed at 0.2 mL/min. A wavelength of 254 nm was detected for MXD.

### 2.5. Evaluation of MXD Formulation Characteristics

Soluble and dissoluble MXD in MXD formulations were separated by centrifugation at 100,000× *g* using a Beckman OptimaTM MAX-XP Ultracentrifuge (Beckman Coulter, Osaka, Japan), and the MXD levels in soluble and dissoluble MXD were analyzed using the HPLC method described as Section 2.4. The MXD formulations were incubated for 2 weeks at 22 °C to evaluate their dispersibility, and samples were taken from 5 mm under the surface over time. The MXD concentrations in the samples were measured using the HPLC method described above. The particle size of MXD-MPs in the range of 0.01–50 µm was measured using the SALD-7100 (Shimadzu Corp., Kyoto, Japan) and the Motic BA210E (Shimazu Corp., Kyoto, Japan), and the particle size of MXD-NPs in the range of 0.01–50 µm and 0–600 nm were measured using the SALD-7100 and the NANOSIGHT LM10 (Quantum Design Japan, Tokyo, Japan), respectively. The refractive index in the SALD-7100 was set at 1.60–0.010 i, and the measurement conditions for the NANOSIGHT LM10 were a viscosity of 1.45 mPa∙s, a wavelength of 405 nm, and a measurement time of 60 s. Additionally, the number of nanoparticles was also measured using the NANOSIGHT LM10. The SPM-9700 was used to obtain an atomic force microscopy (AFM) image of the MXD-NPs (Shimadzu Corp., Kyoto, Japan), and the zeta potential was analyzed using a Zeta Potential Meter Model 502 (Nihon Rufuto Co., Ltd., Tokyo, Japan). The viscosities of the MXD formulations were measured using a Brookfield digital viscometer at 20–40 °C (Brookfield Engineering Laboratories, Inc., Middleboro, MA, USA).

### 2.6. Evaluation of Hair-Growth Effect

The hairs in the dorsal area of the C57BL/6 mice were shaved using an electric razor and electric clippers 2 d before the experiment. The MXD formulations were then applied repetitively to the dorsal area of the C57BL/6 mice once a day, and the subsequent hair growth was monitored daily using a digital camera (at 10:00 a.m.). The changes over time in the area with hair zone was calculated using Image J software. The area under the hair zone–time curve (*AUC*_0–16d_) was analyzed using the Image J data and the trapezoidal rule up. In addition, the C57BL/6 mice were euthanized at 8 d or 10 d after the start of MXD treatment by the injection of a lethal dose of pentobarbital, and the skin was removed; a tissue section of the MXD-treated skin was then prepared. The cross-section was also optically captured using a digital camera.

### 2.7. Quantitative Real-Time Revearse Transcription (RT)-Polymerase Chain Reaction (PCR)

The MXD formulations were applied repetitively to the dorsal area of the C57BL/6 mice once a day, and the mice were euthanized 8 d after the start of the MXD treatments by injection with a lethal dose of pentobarbital; the hair follicles (hair bulge and hair bulb) were removed at 3:00 p.m. The RT and PCR reactions were conducted using an RNA PCR Kit and LightCycler FastStart DNA Master SYBR Green I, and the total amount of RNA in the hair follicle (hair bulge and bulb region) was extracted using the TRIzol reagent and the method (the acid guanidinium thiocyanate–phenol–chloroform extraction method) described in our previous studies [25,34]. The PCR amplification conditions were 95 °C for 10 min, 50 cycles of denaturing (95 °C for 10 s), annealing (66 °C for IGF-1, 64 °C for VEGF and 66 °C or 64 °C for GAPDH for 10 s), and extension (72 °C for 5 s), and the specific primers were as follows: 10 pmol specific primers for IGF-1 (forward 5′-CTGGTCCTGTGTCCCTTTGC-3, reverse 5′-GGACGGGGACTTCTGATC TT-3), VEGF (forward 5′-CAACTTCTGGGCTCTTCTCG-3, reverse 5′-CCTCTCCTCTTCC TTCTCTTC C-3) or glyceraldehyde-3-phosophate dehydrogenase (GAPDH) (forward 5′-TGAAGGTC GGTGTGAACGGATT-3, reverse 5′-CGTGAGTGGAGTCATACTGGAAC-3). The expression levels of IGF-1 mRNA and VEGF mRNA were analyzed from the differences in the threshold cycles for GAPDH and other groups (IGF-1 and VEGF).

### 2.8. Measurement of IGF-1 and VEGF Protein in Hair Follicles

The MXD formulations were applied repetitively to the dorsal area of C57BL/6 mice once a day, and the mice were euthanized at 8 d after the start of MXD treatment by injection with a lethal dose of pentobarbital; the hair follicles (hair bulge and hair bulb) were removed at 3:00 p.m. The IGF-1 levels in the hair follicle were determined using a Mouse/Rat IGF-1 Immunoassay ELISA kit, and the VEGF levels were measured using a VEGF ELISA kit according to our previous study [25]. Briefly, the hair follicle sample (hair bulge and hair bulb region) was homogenized in purified water, and centrifuged at 20,400× *g* for 20 min at 4 °C. The supernatant solution was added into the microplate wells with the pre-coating of monoclonal antibodies specific for mouse IGF-1 or VEGF. The microplates were then incubated at room temperature (IGF-1) or 37 °C under a humid environment (VEGF) for 2 h and washed 4 (IGF-1) and 5 (VEGF) times with Wash Buffer. For the IGF-1 measurement, mouse IGF-1 conjugate regent was added, and the sample were incubated for 2 h at room temperature. After that, the detection reagent was added, and the 30-min incubation of the samples were conducted at room temperature. For the VEGF measurement, both the detection antibody solution regent and the samples were incubated for 1 h at 37 °C in a humidity environment. Then, the plate was washed 4 times using Wash Buffer, and a horseradish peroxidase (HRP)-conjugate antibody was added to the microplates; the samples were then incubated for 1 h at 37 °C in a humidity environment. Microplates with 3,3⌂,5,5⌂-tetramethylbenzidine (TMB) substrate solution for HRP were incubated for 15 min at 37 °C in the dark condition. Both IGF-1 and VEGF were determined to measure the absorbance (450 nm) using a microplate reader.

### 2.9. Measurement of MXD Levels in Skin Tissue, Hair Bulge, Hair Bulb and Blood

The hair in the dorsal area of the C57BL/6 mice was shaved using an electric razor and electric clippers 1 d before the experiment, leaving a length of approximately 2 mm. The skin surface with the 2-mm hair was washed with saline. Thereafter, the MXD formulations were applied to the dorsal area of the C57BL/6 mice and the area was covered with adhesive tape. After 4 h, the C57BL/6 mice were euthanized by injection with a lethal dose of pentobarbital, and samples of the skin tissue, hair bulge, hair bulb and blood were collected. The hairs were pulled by tweezers, and the hair bulge and hair bulb areas were separated; samples obtained from the skin tissue, hair bulge and hair bulb were separately homogenized in ethanol. The homogenates and blood were centrifuged at 20,400× *g* for 20 min at 4 °C, and the supernatants were used to measure the MXD levels. In this study, the MXD contents were measured using the HPLC method described above, and the MXD levels in the skin tissue, hair bulge and hair bulb were represented as mol/mg protein. The protein levels were determined using a Bio-Rad Protein Assay Kit (Bio-Rad Laboratories, Inc., Hercules, CA, USA).

### 2.10. HE and Immunostaining Evaluation of Hair-Growth Effect

The pieces of mouse back skin tissues were fixed in SUPER FIX™ rapid fixative solution (Kurabo Industries, Osaka, Japan) [35]. Paraffin sections were obtained from the fixed mouse back skin tissues in the usual manner and the product certificate and were incubated with anti-CD200 rabbit polyclonal antibody (1:100; Cat No. 14057-1-AP; Proteintech, Rosemont, IL, USA) for 1 h at 37 °C. Both the secondary antibodies and Histofine^®^ Simple Stain™ MAX-PO MULTI (Nichirei, Tokyo) were incubated for 30 min at 37 °C. We used the Liquid 3,3′-Diaminobenzidine Tetrahydrochloride (DAB)+ Substrate Chromogen System (Dako Omnis; Agilent Technologies, Santa Clara, CA, USA) as a colorimetric substrate, and the cell nuclei were stained using hematoxylin. Hematoxylin and eosin (HE) staining was also conducted on continuous sections. The Power BX-51 microscpe (Olympus, Tokyo, Japan) was used for observation.

### 2.11. Statistical Analysis

The data were analyzed using the student *t*-test and Dunnett’s multiple comparison (ANOVA) and expressed as the mean ± standard error (S.E.) of the mean.

## 3. Results

### 3.1. Dispersibility of MXD Formulations

In the previous study, we succeeded in preparing dispersions containing 1% MXD solid nanoparticles using the bead mill method [32,36], and the nanoparticles maintained their dispersed state for 2 weeks. However, an increase in the drug concentration is generally known to accelerate the aggregation of dispersed particles in formulations. Therefore, the dispersibility of MXD nanoparticles needs to be increased. MC is a derivative of cellulose that is highly biocompatible. MC molecules link to water molecules through intermolecular hydrogen bonds to form a cage-like structure, solutions of which are scentless, tasteless and neutral. For these reasons, MC can be used as a thickener and in situ gel base. In addition, MC is useful as an additive for solid nanoparticle drugs, since MC enhances the crushing efficacy in the bead mill. Therefore, the addition of a high MC content may prevent the aggregation of dispersed particles in formulations based on MXD solid nanoparticles. For these reasons, we used 8% MC to prepare the MXD-NPs (suspensions containing 5% MXD solid nanoparticles). We measured the mean particle size, solubility, viscosity and zeta potentials of the MXD formulations (Table 1). Figure 1 shows the particle size distributions and images of the MXD-NPs. The mean particle size of the control MXD-MPs was 5.21 ± 0.93 μm (Figure 1A,B), while the mean particle size of the MXD-NPs was in the nano order with a range of 70–200 nm (Figure 1C,D). The particles in the MXD-NPs appeared as elliptically shaped (Figure 1E) and a nebulous solution (Figure 1F). Figure 2 shows the characteristics of the MXD formulations. The ratio of solid MXD was higher than 99% for the MXD-NPs (less than 1% of the MXD is in solution, Figure 2A), and the ratio of solid MXD in the MXD-NPs was 8.76-fold higher than that in the MXD-MPs. The viscosity of the MXD formulations were enhanced at temperatures near body temperature (40 °C), and no difference was observed in the viscosity and zeta potential between MXD-NPs and MXD-MPs (Figure 2B–D). In addition, aggregation and precipitation were not observed in the MXD-NPs formulation (Figure 2E), and both the ratio of solid MXD in the MXD-NPs formulation and the mean particle size of the MXD-NPs remained unchanged for 14 d at 22 °C (Figure 2F,G).

### 3.2. Effect of MXD Formulations on Hair Growth in C57BL/6 Mice

Next, we investigated the hair-growth effects in C57BL/6 mice when MXD formulations were applied repeatedly (once a day). Hair growth was observed in the mice treated with MXD-NPs or CA-MXD at 8 d or 10 d after the initial application, respectively (Figure 3), and the mice treated with MXD-NPs or CA-MXD showed significant hair growth when compared with the mice treated with the vehicle and MXD-MPs (Figure 4A). The *AUC*_0–16d_ for the MXD-NPs was significantly higher than that of the MXD-MPs and CA-MXD (Figure 4B), and the hair bulbs of the MXD-NPs-treated mice showed enriched growth, compared with that in CA-MXD-treated mice (Figure 4C). Figure 5 shows the mRNA and protein expression levels of IGF-1 and VEGF. These expression levels were significantly higher in mice treated with MXD-NPs or CA-MXD than in those of mice treated with the vehicle and MXD-MPs, and both the IGF-1 and VEGF levels in the mice treated with MXD-NPs were 2.0-fold that of the levels seen in the mice treated with CA-MXD.

### 3.3. Delivery of MXD into the Hair Follicles of C57BL/6 Mice

Figure 6 shows the MXD contents in the skin tissue, hair bulge, hair bulb and blood at 4 h after the administration of the MXD formulations. The MXD concentrations in the skin tissue and hair bulb of the CA-MXD-treated mice were higher than those of the MXD-NPs-treated mice (Figure 6A,C). In contrast, the MXD content in the hair bulge of MXD-NPs-treated mice was significantly enhanced, compared with that in the CA-MXD-treated mice (Figure 6B). Although MXD was detected in blood samples from the CA-MXD-treated mice, MXD was not detected in blood samples from the MXD-NPs-treated mice (Figure 6D).

### 3.4. Proliferation and Avtivation of Hair Follicle Epithelial Stem Cells by MXD Formulations

Figure 7 shows the results of immunostaining with HE and CD200 antibody. Hair follicle growth was observed in all the mice treated with MXD formulations but not in the Vehicle-treated mice (Figure 7A,B). Among them, solid hair follicle growth was observed in the MXD-NPs-treated mice. Moreover, CD200 was observed in all the mice treated with MXD formulations (Figure 7C,D). In particular, CD200 was detected in a limited range around the hair bulge in the CA-MXD-treated mice, whereas CD200 was detected in a wide range between the hair bulge and the sub-bulge in the MXD-NPs-treated mice.

## 4. Discussion

The topical application of MXD is a therapy for AGA, and clinical studies have shown that the efficacy of 5% MXD is significantly higher than that of 1% and 2% MXD [4,5]. In addition, nanomedicines containing MXD are expected to enhance the therapeutic effects of MXD for AGA, improving patient satisfaction in terms of efficacy and safety. However, the design of a MXD solid nanoparticle formulation containing a high (5%) MXD content has not been previously reported. In this study, we developed a new formulation containing 5% MXD nanoparticles that maintain a dispersed state, and we showed that this formulation could be used to deliver MXD effectively into hair follicles (hair bulge), leading the enhancement of hair-growth effects and the activation of the hair follicle epithelial stem cells in C57BL/6 mice.

First, we attempted to design MXD-NPs (a nanoparticle formulation containing a high (5%) MXD content). We previously succeeded in preparing a MXD solid nanoparticle formulation containing a low (1%) MXD content using a bead mill method; the composition of this formulation was 1% MXD, 2% MC, 0.5% mannitol, 0.026% methyl p-hydroxybenzoate and 0.014% propyl p-hydroxybenzoate [25]. In the present study, we prepared a formulation containing 5% MXD using the same process and compositions; however, the bead mill process resulted in the MXD reaching “a meringue state” (a mixture of MXD, water and air). This meringue state can be improved by changing the MC content [32]. Therefore, we performed bead mill treatments using MC contents of 4%, 6%, 8% and 10%. Although a meringue state was obtained by mill treatments using 4% and 6% MC, the addition of 8% and 10% MC solved the problem. On the other hand, MC is commonly used as a thickener and in situ gel base, and MXD formulations using 8% and 10% MC were in a liquid state and an almost gel-like state, respectively (viscosity of MXD formulation containing 10% MC at 30 °C, 76.2 ± 2.9 mPa∙s). Therefore, we decided to use an 8% MC content, since a liquid state is more suitable for AGA treatments than a gel-like state.

Next, we measured the characteristics of the nanoparticle formulation containing a high MXD content that was prepared in this study. The particle sizes of the MXD-NPs were 70–200 nm (mean particle size, 139.8 ± 8.9 nm, Figure 1D), and the ratio of solid MXD in the MXD-NPs was higher than 99%. Generally, increasing the drug concentration can accelerate the aggregation of dispersed particles in formulations. However, no aggregation was detected in the MXD-NPs formulation after 14 days at 22 °C, and no change in the ratio of solid MXD was seen (Figure 2). The viscosity of MXD-NPs was approximately 38 mPa∙s at 30 °C and 144 mPa∙s at 40 °C, and the high viscosity produced by 8% MC content may have enhanced the dispersibility of the MXD nanoparticles.

Furthermore, we evaluated hair-growth effects of MXD-NPs using C57BL/6 mice. The MXD-NPs-treated mice showed hair growth more rapidly than the CA-MXD-treated mice, and the *AUC*_0–16d_ for the MXD-NPs group was significantly greater than the *AUC*_0–16d_ for the CA-MXD group (Figure 3 and Figure 4). The mRNA and protein expression levels of hair-growth factors were also higher in the MXD-NPs-treated mice than in the CA-MXD-treated mice (Figure 5). These results indicated that the hair-growth effect of the MXD-NPs was superior to that of CA-MXD. Moreover, we newly evaluated the pathway for hair follicle delivery dividing the hair follicle into the upper part (hair bulge) and the lower part (hair bulb) in addition to the evaluation of the MXD content in the skin tissue and blood. The MXD concentration in the hair bulge of the MXD-NPs-treated mice was higher than that of the CA-MXD-treated mice, whereas the contents in the hair bulb and the skin tissue of the MXD-NPs-treated mice were lower than those of CA-MXD-treated mice (Figure 6). In addition, MXD was not detected in the blood of the MXD-NPs-treated mice (Figure 6). Drugs are generally known to penetrate the skin across the continuous stratum corneum (SC) or via the appendages [37]. Nagai et al. reported that the SC prevents the penetration of nanoparticles with a mean diameter of around 200 nm [34]. Drugs dissolved in formulations are considered to permeate the SC and to penetrate deeper into the skin tissue, as well as the bloodstream. In the present study, the particle sizes of the MXD-NPs were 70–200 nm, and these nanoparticles might have difficulty passing through the SC. The ratio of solid MXD in the MXD-NPs was over 99%, meaning that the concentration of dissolved MXD in the vehicle was less than 0.05% (Figure 2A). Therefore, only a small amount of MXD would have been able to permeate the SC. This might explain the low MXD contents in the skin tissue and blood of the MXD-NPs-treated mice. Alternatively, the nanoparticles might migrate into the hair follicles through the pores [25], enhancing the therapeutic effects of MXD for the treatment of AGA and improving patient satisfaction in terms of efficacy and safety. In our previous study examining a 1% MXD nanoparticle formulation, the formulation increased the MXD content in the hair bulb, compared with a commercially available product (1% MXD solution) [25]. On the other hand, the MXD content in the hair bulge of the MXD-NPs-treated mice was higher than that of the CA-MXD-treated mice, although the MXD content in the hair bulb of the MXD-NPs-treated mice was lower than that of the CA-MXD-treated mice (Figure 6). This result for the hair bulb contradicts the present result. A high viscosity is known to attenuate the fluidity of suspensions, and the viscosity of 5% MXD-NPs was 31-fold that of the 1% MXD-NPs formulation (1.21 mPa∙s) at 30 °C (Table 2). Therefore, the viscosity of the MXD-NPs may be responsible for this contradiction. Further study is needed to clarify the details of the relationship between viscosity and the skin penetration of MXD-NPs.

In the present study, the MXD-NPs formulation produced hair growth earlier than the CA-MXD formulation (Figure 3 and Figure 4A), and the MXD content was higher in the hair bulge of the MXD-NPs-treated mice than that in the CA-MXD-treated mice (Figure 6B), whereas the MXD content was lower in the hair bulb of the MXD-NPs-treated mice than in the CA-MXD-treated mice (Figure 6C). These findings indicated that the MXD content in the hair bulge may be more strongly correlated with hair-growth effects, compared with the MXD content in the hair bulb. The cells in the bulge region are known to possess stem cell characteristics and to generate secondary germ cells, which produce new hair shafts at anagen onset [38]. We hypothesized that the MXD delivered into the hair bulge activated hair follicle epithelial stem cells. Therefore, we performed immunohistochemical staining to evaluate the proliferation and activation of stem cells expressing CD200. Our results showed that the expression of CD200 was observed in the bulge and sub-bulge regions in the MXD-NPs-treated mice, while CD200 was observed in a more limited region of the hair bulge in the CA-MXD-treated mice (Figure 7). Taken together, these results suggest that MXD-NPs deliver a high MXD content into the hair bulge, which may lead to the proliferation and activation of hair follicle epithelial stem cells and the enhancement of hair-growth effects (Figure 8). However, the present understanding of how MXD stimulates hair growth is limited, and little is known about direct interactions between MXD and stem cells. Therefore, further study of whether MXD acts directly on stem cells is now being planned.

## 5. Conclusions

We succeeded in designing a new formulation containing 5% MXD nanoparticles, and the dispersibility of the MXD nanoparticles was maintained for at least 14 days at 22 °C. Our results showed that the MXD-NPs delivered MXD into the hair follicle (hair bulge) more effectively than CA-MXD and that the therapeutic efficacy of MXD-NPs for hair growth was higher than that of CA-MXD. Moreover, MXD-NPs may activate hair follicle epithelial stem cells. These promising findings may lead to the therapeutic effects of MXD formulations satisfying AGA patients in terms of efficacy and safety.

## Figures and Tables

**Figure 1 pharmaceutics-14-00947-f001:**
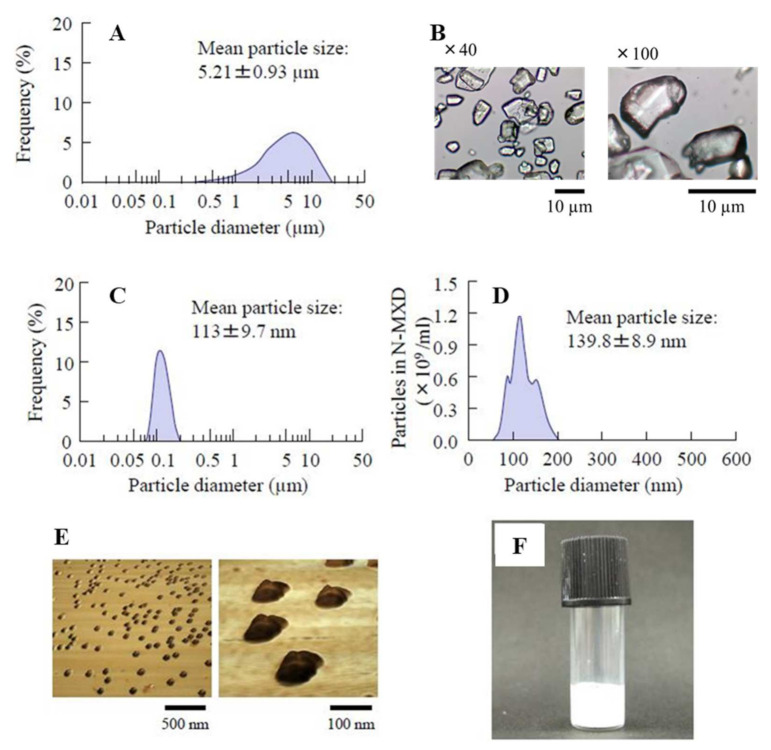
Distribution of MXD particle sizes with and without a bead mill. (**A**) The particle size distributions for MXD-MPs without a bead mill, and (**B**) Images of MXD-MPs without a bead mill. The data were measured using the SALD-7100 and the Motic BA210E, respectively. (**C**,**D**) The particle size distributions for MXD-NPs with a bead mill. The data were measured using the SALD-7100 and NANOSIGHT LM10, respectively. (**E**) AFM images of MXD-NPs with a bead mill. (**F**) Digital photo of MXD-NPs with a bead mill.

**Figure 2 pharmaceutics-14-00947-f002:**
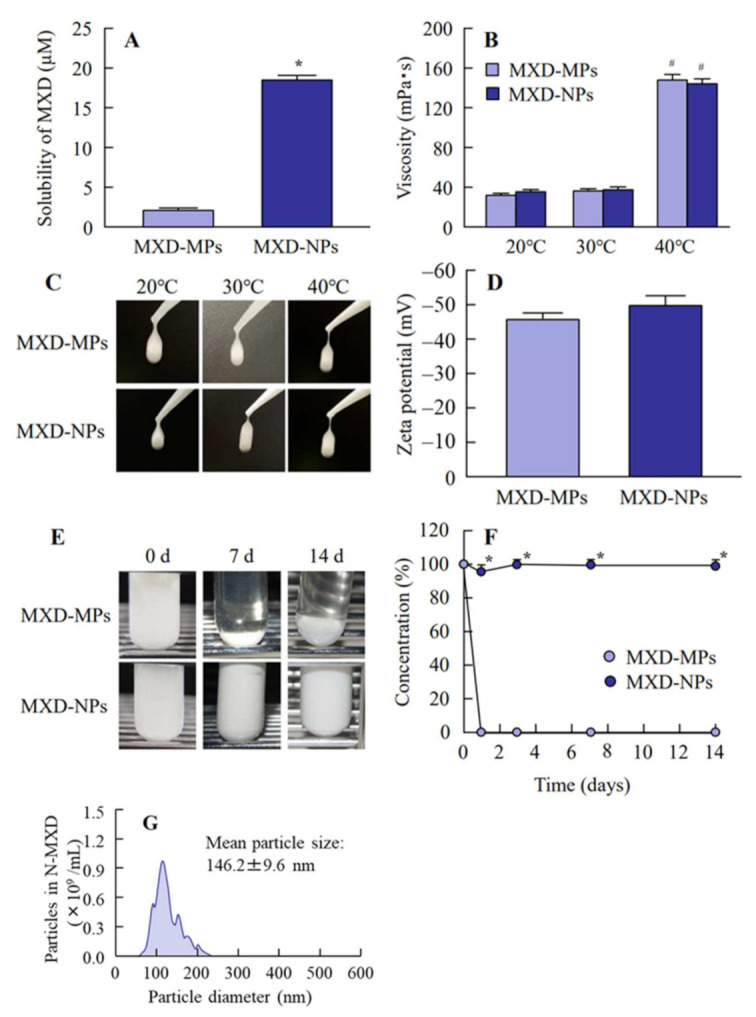
Solubility, viscosity, zeta potential and dispersibility of MXD-MPs and MXD-NPs. (**A**) Solubility of MXD in MXD-MPs and MXD-NPs. (**B**) Viscosity of MXD-MPs and MXD-NPs. (**C**) Images of MXD-MPs and MXD-NPs at 20–40 °C. (**D**) Zeta potential of MXD-MPs and MXD-NPs, (**E**) and (**F**) Images (**E**) and dispersibility (**F**) of MXD-MPs and MXD-NPs. (**G**) Particle size distribution of MXD-NPs after the 14-d incubation. *n* = 7. * *p* < 0.05 vs. MXD-MPs. ^#^ *p* < 0.05 vs. MXD-MPs at 20 °C for each category. The solubility of the MXD-NPs was 8.76-fold that of the MXD-NPs, and over 99% of the MXD was solid MXD in the MXD-NPs. The viscosity was enhanced at 40 °C, and no aggregation was observed for 14 d in the MXD-NPs.

**Figure 3 pharmaceutics-14-00947-f003:**
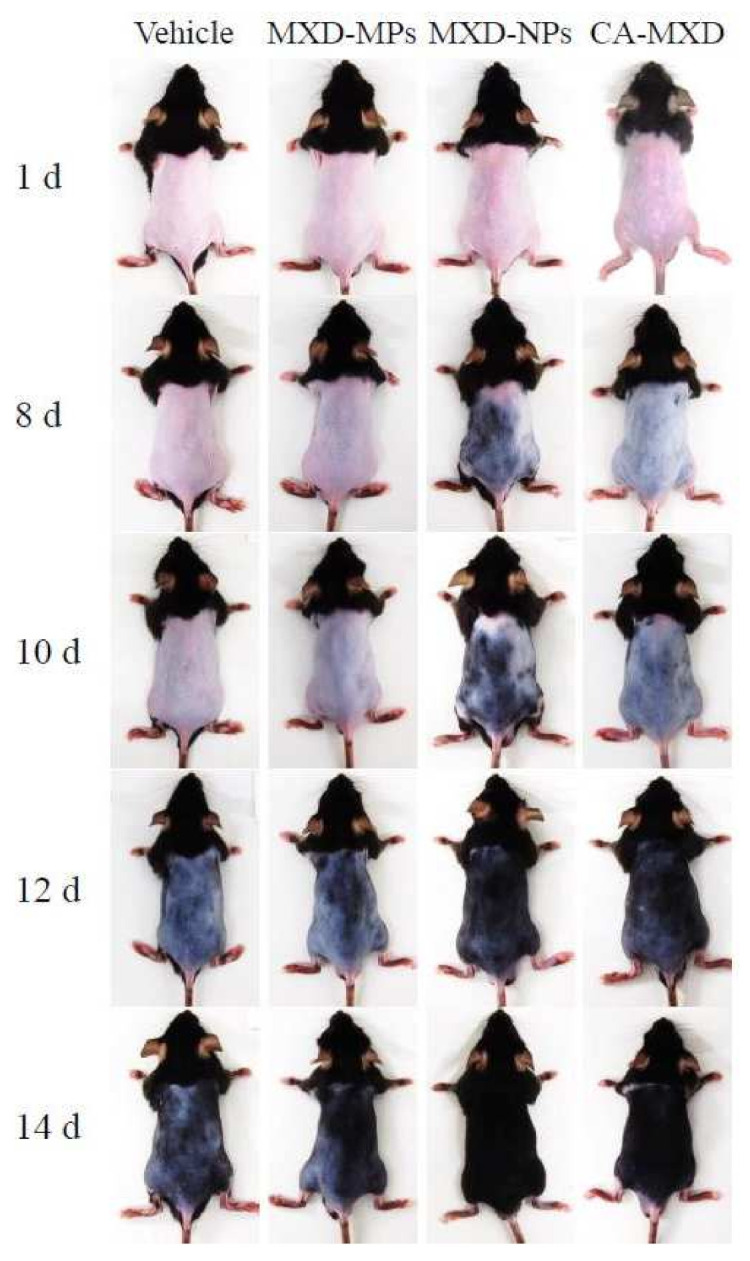
Digital photos of C57BL/6 mice repetitively treated with MXD formulations (once a day). Vehicle, only the vehicle used to prepare the MXD-MPs and MXD-NPs formulations was applied. MXD-MPs, MXD-MPs-applied mice. MXD-NPs, MXD-NPs-applied mice. CA-MXD, CA-MXD-applied mice. The repetitive applications of MXD-NPs enhanced the hair growth, compared with repetitive applications of MXD-MPs or CA-MXD.

**Figure 4 pharmaceutics-14-00947-f004:**
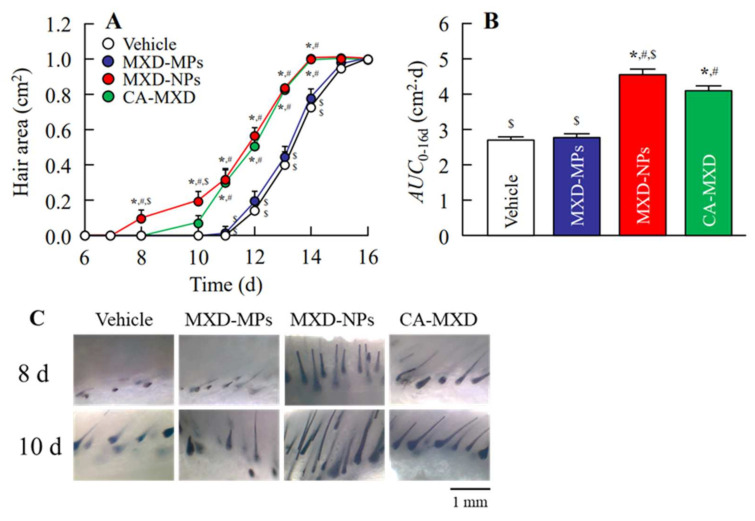
Effect of the repetitive application of MXD formulations on hair growth in C57BL/6 mice. (**A**) Changes in hair area and (**B**) *AUC*_0–16d_ in C57BL/6 mice treated with repeated applications of MXD formulations (once a day). (**C**) Image of the skin cross-section of C57BL/6 mice skin after 8 d and 10 d of repeated applications of the MXD formulations (once a day). Vehicle, only the vehicle used to prepare the MXD-MPs and MXD-NPs formulations was applied. MXD-MPs, MXD-MPs-applied mice. MXD-NPs, MXD-NPs-applied mice. CA-MXD, CA-MXD-applied mice. *n* = 6–10. * *p* < 0.05 vs. Vehicle for each group. ^#^
*p* < 0.05 vs. MXD-MPs for each group. ^$^
*p* < 0.05 vs. CA-MXD for each group. The *AUC*_0–16d_ of the MXD-NPs-applied and CA-MXD-applied mice were significantly higher than that of the Vehicle and MXD-MPs groups. Moreover, hair growth started at 8 and 10 days after the initial treatment with MXD-NPs and CA-MCD, respectively, and hair-growth effects at 8–10 d after treatment were only observed in mice treated with MXD-NPs.

**Figure 5 pharmaceutics-14-00947-f005:**
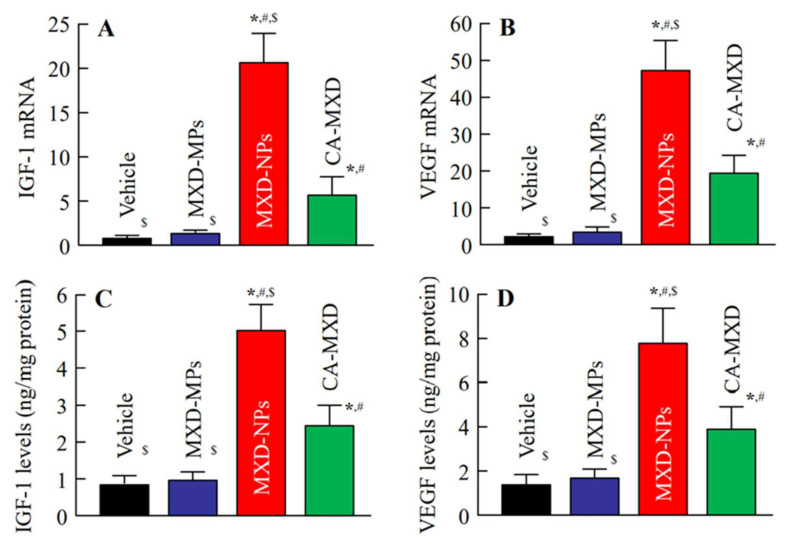
Effect of the MXD formulations (particle types and solution type) on IGF-1 and VEGF in the hair follicles of C57BL/6 mice at 8 d after the start of repetitive applications (once a day). (**A**) Expression of IGF-1 and (**B**) VEGF mRNA in C57BL/6 mice treated with MXD formulations. (**C**) IGF-1 and (**D**) VEGF levels in C57BL/6 mice treated with MXD formulations. The samples were prepared from the hair follicle (hair bulge and hair bulb region). Vehicle, only the vehicle used to prepare the MXD-MPs and MXD-NPs formulations was applied. MXD-MPs, MXD-MPs-applied mice. MXD-NPs, MXD-NPs-applied mice. CA-MXD, CA-MXD-applied mice. *n* = 5–9. * *p* < 0.05 vs. Vehicle for each category. ^#^
*p* < 0.05 vs. MXD-MPs for each group. ^$^
*p* < 0.05 vs. CA-MXD for each group. The repeated treatment of MXD-NPs increased both the mRNA and protein levels of IGF-1 and VEGF, compared with those in the Vehicle, MXD-MPs and CA-MXD groups.

**Figure 6 pharmaceutics-14-00947-f006:**
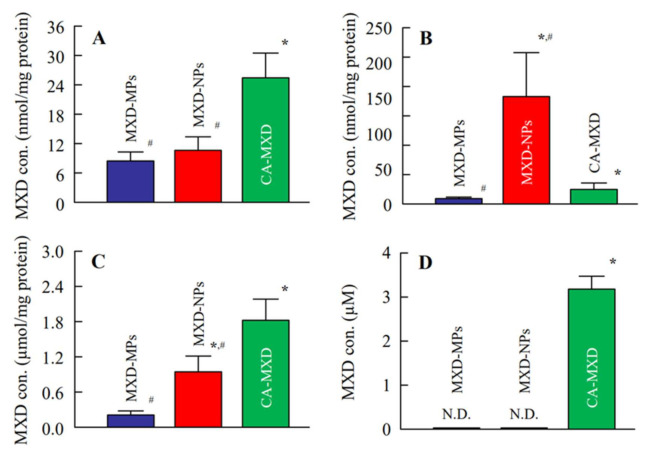
MXD contents in the skin tissue (**A**), hair bulge (**B**), hair bulb (**C**) and blood (**D**) of C57BL/6 mice 4 h after the application of MXD formulations. *n* = 4–12. MXD-MPs, MXD-MPs-applied mice. MXD-NPs, MXD-NPs-applied mice. CA-MXD, CA-MXD-applied mice. N.D., not detectable. * *p* < 0.05 vs. MXD-MPs for each group. ^#^
*p* < 0.05 vs. CA-MXD for each group.

**Figure 7 pharmaceutics-14-00947-f007:**
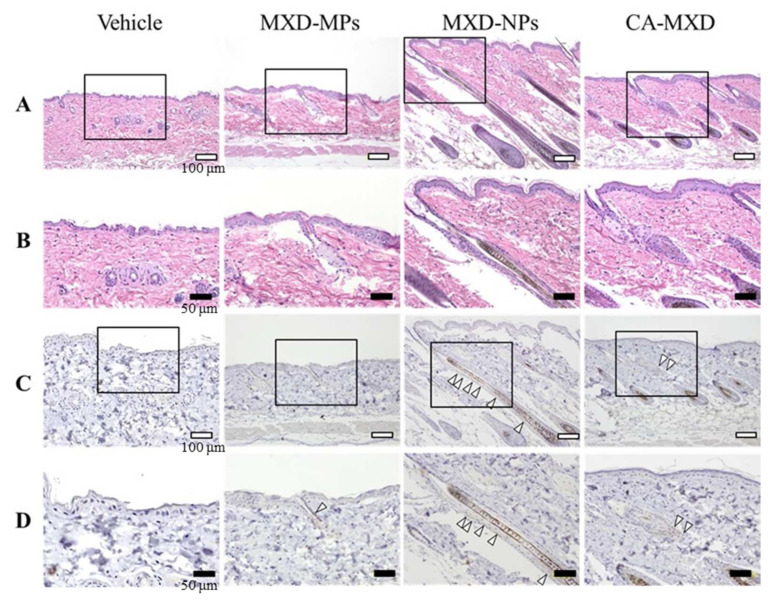
Microscopic effects of MXD formulations on skin tissue (hair bulge and hair bulb) of C57BL/6 mice at 8 d after the start of repetitive applications (once a day). (**A**) Images of H&E-stained skin tissue specimens from C57BL/6 mice (bars indicate 100 µm). (**B**) High magnification microscope images in the areas designated by the squares in Figure A (bars indicate 50 µm). (**C**) Optical images of immunostaining for CD200 in the serial segments shown in Figure A (bars indicate 100 µm). (**D**) High magnification microscope images of the areas designated by the squares in Figure C (bars indicate 50 µm). Vehicle, only the vehicle used to prepare the MXD-MPs and MXD-NPs formulations was applied. MXD-MPs, MXD-MPs-applied mice. MXD-NPs, MXD-NPs-applied mice. CA-MXD, CA-MXD-applied mice. The arrow shows the expression of CD200. CD200 is stained brown; the expression of CD200 was enhanced in the MXD-NPs group, compared with the CA-MXD group.

**Figure 8 pharmaceutics-14-00947-f008:**
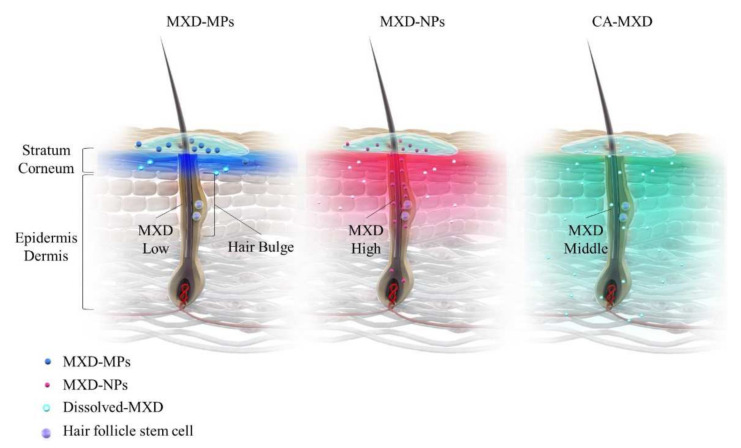
Schemes for drug delivery into the hair bulge and bulb region for MXD-MPs (powder type), MXD-MPs (NPs type) and CA-MXD (solution type) formulations.

**Table 1 pharmaceutics-14-00947-t001:** Characteristics of MXD-MPs and MXD-NPs.

Formulation	Mean Particle Size (μm)	Solubility (μm)	Ratio of Solid MXD	Viscosity (mPa∙s)		Zeta Potentials (mV)
				30 °C	40 °C	
MXD-MPs	5.21 ± 0.93	2.11 ± 0.61	More than 99%	36	148	–46.7
MXD-NPs	0.14 ± 0.009	18.5 ± 0.68	More than 99%	38	144	–61.5

**Table 2 pharmaceutics-14-00947-t002:** Differences between 5% MXD-NPs and 1% MXD-NPs.

Formulation	Mean Particle Size (nm)	Ratio of Solid MXD	Viscosity at 30 °C (mPa∙s)	MXD Contents versus a Commercially Available MXD Solution (CA-MXD)			Hair Growth Effect versus CA-MXD
				Hair Bulge	Hair Bulb	Skin∙Blood	
5% MXD-NPs	139.8 ± 8.9	More than 99%	38	↑	↓	↓	↑
1% MXD-NPs	153.0 ± 3.8	61.2%	1.21	-	↑	↓	↑

“↑” means that higher than that of CA-MXD, “↓” means that lower than that of CA-MXD.

## Data Availability

Not applicable.

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
