# Peer review of "Minoxidil Nanoparticles Targeting Hair Follicles Enhance Hair Growth in C57BL/6 Mice"

_pharmaceutics, 2022, doi:10.3390/pharmaceutics14050947_

Round 1

Reviewer 1 Report

The authors present a well-designed and well-written study.  The experimental work is described in appropriate detail.  My main question concerns the animal model - although the results obtained with the C57BL/6 mice are quite striking and promising - how predictive is this model of behaviour in humans ? The manuscript could be improved by adding some discussion of the relevance of the model and potential challenges for translation to humans.  

Author Response

We carefully revised our manuscript acording to the suggestions of the reviewer 1, and details are as follows.

Point 1: My main question concerns the animal model - although the results obtained with the C57BL/6 mice are quite striking and promising - how predictive is this model of behaviour in humans ? The manuscript could be improved by adding some discussion of the relevance of the model and potential challenges for translation to humans.

Response 1: Thank you very much for pointing this out. The stump-tailed macaque is known as an animal that develops androgenetic alopecia and is used as an animal model to translate hair-growth effect to humans [1]. Besides, hair-growth test using mice, observation of human hair follicles organ and proliferation analysis of human dermal papilla cells are generally conducted from the viewpoint of the convenience for evaluation of hair-growth effects [2]. On the other hand, there are differences between human and mice hair cycle: the human hair cycle is mosaic, while the mice hair cycle is synchronized [3]. In regard to minoxidil, the hair-growth efficacy and the concentration dependency are verified in human [4], stump-tailed macaque [5] and mice [6]. In this regard, we used C57BL/6 mice as an animal model. We added some discussion on this point (line 70-78).

[Refference]

  1. Uno, H. The stumptailed macaque as a model for baldness: effects of minoxidil. Int. J. Cosmet. Sci. 1985, 8, 63-71.
  2. Zhang, Y.; Ni, C.; Huang, Y. et al. Hair growth-promoting effect of resveratrol in mice, human hair follicles and dermal papilla cells. Clin. Cosmet. Invest. Dermatol. 2021, 14, 1805-1814.
  3. Oh, J.W.; Kloepper, J.; Langan, E.A. et al. A guide to studying human hair follicle cycling in vivo. J. Invest. Dermatol. 2016, 136, 34-44.
  4. Olsen, E.A.; Dunlap, F.E.; Funicella, T. et al. A randomized clinical trial of 5% minoxidil versus 2% topical minoxidil and placebo in the treatment of androgenetic alopecia in men. J. Am. Acad. Dermatol. 2002, 47, 377-385.
  5. Uno, H.; Cappas, A.; Bringham, P. Action of topical minoxidil in the bald stump-tailed macaque. J. Am. Acad. Dermatol. 1987, 16, 657-668.
  6. Takeshita, K.; Yamagishi, I.; Sugimoto, T. et al. The hair growing effect of minoxidil. Ann. NY Acad. Sci. 1991, 642, 470-472.

Thank you for your great comments.

Reviewer 2 Report

Reviewer’s comments

The article entitled “Minoxidil nanoparticles targeting hair follicles enhance hair 2 growth in C57BL/6 mice” described the impact of MXD-NPS (obtained in present study) and MXD (commercial) on hair-growth using C57BL/6 mice. Additionally the authors investigate effect of MXD-NPs on the production 85 of IGF-1 and VEGF. The study is well described, the complimentary is also achieved. However, few aspects should be specified and explain.

  1. Fig 1: please clarify which results are for MXDNPs and which for MXDMPs. When it is presented such characteristics should be shown comparison between them, especially the size and stability (zeta). AFM results also is presented only for one type of particles. For better interpretation should be performed for both investigated particles (NPs/MPs). Additionally, using AFM software also it is possible to identified predominant population in the sample (regarding the size); this aspect will prove the discussion regarding the zeta potential parameter. This aspect is crucial once the stability of the system it is correlated with the size and as a consequence explain the bio viability of the particles. Higher size – low stability – low distribution and vice versa. Moreover, AFM should be also curry out for MXD before and after melding process. Furthermore, from the (D) we cannot see spherical form of NPs, due to the fact that firstly, you have coated material with lower size that is able to adhere to each other forming bigger size of particles (as we can see from picture). 3D version of the results should be also taken in consideration.      
  2. Fig 2. The author described very well the comparison between NPs and MPs. However, the authors have been written that no aggregation was observed. how the authors deduced this conclusion? Based only on organoleptic properties (E)? to confirm no aggregation or changing the size and/or stability should be performed size, zeta measurement during the 14 days. Drastic changes could not appeared but slight differences should be. Also appearance of additional signal with higher size distribution value could (not must) be. After, the authors can prove the stability of the system after 14 days.   

Very well authors described further results showing the comparison. However, poor discussion is presented concerning  the size and stability aspect in context of  bio viability and delivery properties.      

Author Response

Response to Reviewer 2 Comments

We carefully revised our manuscript acording to the suggestions of the reviewer 2, and details are as follows.

Point 1: Fig 1. Please clarify which results are for MXDNPs and which for MXDMPs. When it is presented such characteristics should be shown comparison between them, especially the size and stability (zeta).

Response 1: Thank you very much for pointing this out. We revised the caption of Fig 1 in order to clarify which results are for MXD-NPs and which for MXD-MPs (line 293-299). Table 1 shows the characteristics of MXD-MPs and MXD-NPs such as the size, zeta potentials and viscosity (line 290).

Point 2: AFM results also is presented only for one type of particles. For better interpretation should be performed for both investigated particles (NPs/MPs). Additionally, using AFM software also it is possible to identified predominant population in the sample (regarding the size); this aspect will prove the discussion regarding the zeta potential parameter. This aspect is crucial once the stability of the system it is correlated with the size and as a consequence explain the bio viability of the particles. Higher size – low stability – low distribution and vice versa.

Response 2: We thank the reviewer for this pertinent suggestion. We attempted to measure both particles (MXD-NPs/MXD-MPs) using AFM, however, AFM was not suitable to measure the MXD-MPs, since the particle size of MXD-MPs was too large. For the reaseon, we performed microscopic observation of MXD-MPs using the Motic BA210E (Shimazu Corp., Kyoto, Japan) and the results are shown in the Figure A. below. We added the results to Figure 1B. (line 292).

Figure A. Microscopic observation of MXD-MPs (the particles without a bead mill) using the Motic BA210E.

Point 3: Moreover, AFM should be also curry out for MXD before and after melding process.

Response 3: The reviewer’s comment is very important. In the previous study [1], we have confirmed that there is no change in the crystallinity of MXD before and after applying the bead mill method, and the results are shown in the Figure B. below.

Figure B. Morphology of MXD with or without bead mill treatment in XRD. (A) XRD pattern of MXD not treated with bead mill. (B) XRD pattern of MXD treated with bead mill.

[Refference]

  1. Nagai, N.; Iwai, Y.; Sakamoto, A. et al.; Drug delivery system based on minoxidil nanoparticles promotes hair growth in C57BL/6 mice. Int. J. Nanomedicine. 2019, 14, 7921-7931.

Point 4: Furthermore, from the (D) we cannot see spherical form of NPs, due to the fact that firstly, you have coated material with lower size that is able to adhere to each other forming bigger size of particles (as we can see from picture). 3D version of the results should be also taken in consideration.

Response 4: The reviewer’s comments are correct. We revised from “distorted spheres” to “elliptically shaped” (line 277-278).

Point 5: Fig 2. The author described very well the comparison between NPs and MPs. However, the authors have been written that no aggregation was observed. how the authors deduced this conclusion? Based only on organoleptic properties (E)? to confirm no aggregation or changing the size and/or stability should be performed size, zeta measurement during the 14 days. Drastic changes could not appeared but slight differences should be. Also appearance of additional signal with higher size distribution value could (not must) be. After, the authors can prove the stability of the system after 14 days.

Response 5: We appreciate the reviewer’s comment on this point. In order to confirm no aggregation, in addition to the observation of the appearance and the measurement of the MXD content in the supernatant solution (as shown in Figure 2.E and 2F in line 302), we measured the particle size distribution after 14 days of incubation. Figure C. below shows the results of the particle size distributions before and after the 14 days incubation, and we found that no significant change in the mean particle size was observed. We added the result to Figure 2G (line 302).

Figure C. The mean particle size distribution of MXD-NPs before (A) and after (B) the 14 days incubation.

Thank you for your great comments.

Reviewer 3 Report

In this manuscript, Oaku et al describe a 5% minoxidil formulation in methylcellulose nanoparticles that provides superior properties and superior promotion of hair regrowth in a C67BL/6 mouse model, compared to (non-milled) minoxidil microparticles and compared to commercial 5% minoxidil.

The paper is straightforward, logical, easy to read, compelling, and performs the correct experiments to make the conclusions that it wants to make. The only comment I have to be fixed is that, in 2.11, authors say that they used ANOVA and Dunnett’s multiple comparisons test for statistical comparisons of more than two groups. But in figures 5 and 6, authors give comparisons of each group to multiple other groups. Does this mean that the authors applied ANOVAs and Dunnett’s test consecutively in order to compare each group against a single group? (e.g. Dunnett’s test for comparison to vehicle, then Dunnett’s test for comparison to CA-MXD, etc.). If so, wouldn’t it make more sense just to use Tukey’s test or the Holm-Sidak method, which already have built in corrections for comparisons across all groups?

Author Response

Response to Reviewer 3 Comments

We carefully revised our manuscript acording to the suggestions of the reviewer 3, and details are as follows.

Point 1: The only comment I have to be fixed is that, in 2.11, authors say that they used ANOVA and Dunnett’s multiple comparisons test for statistical comparisons of more than two groups. But in figures 5 and 6, authors give comparisons of each group to multiple other groups. Does this mean that the authors applied ANOVAs and Dunnett’s test consecutively in order to compare each group against a single group? (e.g. Dunnett’s test for comparison to vehicle, then Dunnett’s test for comparison to CA-MXD, etc.). If so, wouldn’t it make more sense just to use Tukey’s test or the Holm-Sidak method, which already have built in corrections for comparisons across all groups?

Response 1: The reviewer’s comment is correct.As you mentioned, Tukey’s test and the Holm-Sidak method are useful test method for multiple comparisons. Dunnett’s test is also applicable for multiple comparisons, so in the present study, we used Dunnett’s test consecutively. Thank you very much for pointing this out.

Thank you for your great comments.

Reviewer 4 Report

Based on a previous research paper from 2019, the authors further improved solid particle topical therapy capable of nanodelivering Minoxidil (MXD-NPs) and stimulating hair growth at the hair bulge level. The work is solid and well structured, and the experimental section performed using a preclinical in vivo model (C57BL/6 mice) is also consistent. In this study they increased the amount of the active component in the formulation to 5% (compared to the previous 1%), with the aim of achieving both the maintenance of the dispersion state and the enhancement of the effects of hair growth. The design of a MXD solid nanoparticle formulation containing a high MXD content (5%) has not been previously reported, and this makes the work of particular interest. The experimental study for the characterization of the obtained nanoparticles is convincing and even at 5% of MXD the formulation works well with some advantages over the previous one. Table 2 and Figure 8 summarize these concepts very well. As well as including a commercially available MXD (CA-MXD), I believe it may be useful to include in biological experiments demonstrating the effectiveness of the 5% formulation also some experimental data referring to MXD-NPs at 1%. The authors could integrate (if these data are available or use data from previously published paper) information about the effects of 1% solid particles in figures 4 and 5. It would be nice, for example, to show how the more concentrated formulation is even more effective both in the hair bulb stimulation and in the regulation of IGF-1 and VEGF expression.  

Minor:

- lane 70:  change to “animal model”

- 380-384 lanes: add ref 25 in this section of discussion

Author Response

Response to Reviewer 4 Comments

We carefully revised our manuscript acording to the suggestions of the reviewer 4, and details are as follows.

Point 1: Table 2 and Figure 8 summarize these concepts very well. As well as including a commercially available MXD (CA-MXD), I believe it may be useful to include in biological experiments demonstrating the effectiveness of the 5% formulation also some experimental data referring to MXD-NPs at 1%. The authors could integrate (if these data are available or use data from previously published paper) information about the effects of 1% solid particles in figures 4 and 5. It would be nice, for example, to show how the more concentrated formulation is even more effective both in the hair bulb stimulation and in the regulation of IGF-1 and VEGF expression.

Response 1: Thank you very much for pointing out this point. In the present study, we could not integrate and make direct discussion the differences between the effects of 1% and 5% solid nanoparticles in figure 4 and 5 because, in the animal model, baselines of hair growth scores would vary across studies and the groups to be compared must be evaluated at the same time. We would like to consider a direct comparison of 1% and 5% solid nanoparticles as a further study.

Point 2: Minor:

- lane 70:  change to animal model

- 380-384 lanes: add ref 25 in this section of discussion

Response 2: We revised as you pointed out (line 79 and 411-415).

Thank you for great comments.

Round 2

Reviewer 2 Report

Thank you very much for the answers.

I have only one additional remark. I am not satisfied from the microscope image for MPs. I understood that particles have bigger size however, AFM should be performed not at 500nm but higher (for ex 10 µm or 5 or 1 µm), should be shown the differences between surface in AFM, if compare this.  

Author Response

Response to Reviewer 2 Comments

We carefully revised our manuscript acording to the suggestions of the reviewer 2, and details are as follows.

Point 1: I have only one additional remark. I am not satisfied from the microscope image for MPs. I understood that particles have bigger size however, AFM should be performed not at 500nm but higher (for ex 10 µm or 5 or 1 µm), should be shown the differences between surface in AFM, if compare this.

Response 1: Thank you very much for providing these insights. We have performed AFM for MXD-MPs in a wide view and the result is shown in the Figure below. We have confirmed the particle size of MXD-MPs, but not the surface shape since it was difficult to set conditions for imaging large particles. Therefore, we don’t reflect the data in the present manuscript. We will demonstrate the measurement conditions for AFM image of MXD-MPs, and indicate them in a future report. Thank you very much for pointing this out.

Figure. AFM image of MXD-MPs without a bead mill.

Thank you for your great comments.
